# Effects of Childcare Hands-On Gardening on Preschoolers’ (3–5 Years) Physical Activity in Semi-Arid Climate Zone

**DOI:** 10.3390/ijerph21050548

**Published:** 2024-04-26

**Authors:** Muntazar Monsur, Tristen Hefner, Jason Van Allen, Nazia Afrin Trina, S. Y. Andalib, Nilda Cosco

**Affiliations:** 1Department of Landscape Architecture (DoLA), Davis College of Agricultural Sciences and Natural Resources, Texas Tech University, Lubbock, TX 79409, USA; ntrina@ttu.edu (N.A.T.); sandalib@ttu.edu (S.Y.A.); 2Department of Psychological Sciences, Texas Tech University, Lubbock, TX 79409, USA; tristen.hefner@ttu.edu (T.H.); jason.vanallen@ttu.edu (J.V.A.); 3Department of Landscape Architecture and Environmental Planning, College of Design, North Carolina State University, Raleigh, NC 27695, USA; ngcosco@ncsu.edu

**Keywords:** gardening, childcare, physical activity, accelerometers, preschool-aged children, obesity, semi-arid climate

## Abstract

How hands-on gardening impacts behaviors including healthy eating and physical activity during early childhood can be of critical importance for preventing the early onset of obesity. This study investigates how participating in hands-on gardening impacts preschoolers’ (3–5 years old) physical activity (measured by accelerometers) in childcare centers in the semi-arid climate zone. The research was conducted in eight licensed childcare centers located in West Texas with 149 children (*n* = 149). Four childcare centers in the experimental group received hands-on garden interventions; the other four in the control group did not. In both experimental (intervention) and control (non-intervention) centers, children wore Actigraph GT3X+ accelerometers continuously for 5 days before and for 5 days after intervention (a total of 10 days). Results show that the duration of sedentary behavior of children in the experimental (intervention) group significantly decreased compared to children in the control (non-intervention) group. The finding suggests that the positive effects of childcare hands-on gardening on physical activity extend to semi-arid climate zones where gardening is challenging due to high temperatures and lack of annual rainfall. The research emphasizes the critical need to incorporate hands-on gardening in childcare centers as an obesity prevention strategy nationally in the US and beyond.

## 1. Introduction

Childhood obesity is a serious problem in the United States, putting children and adolescents at risk for poor health. A growing body of research suggests that the risk for childhood and adolescent obesity appears during the early years. Children with a high BMI level or who are overweight or obese in their early years are more likely to be overweight or obese in the later years of their lives [1,2]. For US children and adolescents aged 2–19 years in 2017–2020, the prevalence of obesity was 19.7% with an obesity prevalence of 12.7% among 2 to 5 year olds [3]. Early childhood has been identified as a critical period for developing interventions to promote healthy eating preferences and physical activity (PA) patterns [4,5]—two major protective/preventive behaviors for the prevention of the early onset of obesity.

Intervention-based obesity prevention techniques are found to be effective in promoting children’s PA in the early years. Hands-on gardening, which is found to be associated with higher levels of PA for adults [6,7] and school-agers [8,9] can also be an effective intervention-based obesity prevention technique during early childhood. Two recent studies conducted in North Carolina (NC) [10,11] found positive associations between hands-on gardening and both PA and healthy eating preferences for preschoolers attending center-based childcare. These are among the very few studies that investigated the potential role of gardening in licensed center-based childcare facilities for preventing obesity during the early years of childhood. However, more research is needed to understand whether such positive associations (between hands-on gardening and PA) would extend in different climate zones—especially in arid or semi-arid climates where fruit and vegetable (FV) gardening is more challenging compared to a humid subtropical climate zone like NC. The success of FV gardens in terms of plant growth and yield largely depend on climate conditions and environmental factors including annual rainfall, humidity, annual sunshine hours, soil moisture, etc. The effectiveness of FV hands-on gardening as a health intervention, therefore, may not be the same in different US areas with varying gardening conditions. For instance, West Texas (WT) receives only about 20 inches of rain annually, while NC receives about 48 inches (more than double). The average annual relative humidity for these two places is also significantly different (WT 44.5%, NC 71%). On the flip side, WT enjoys 264 sunny days a year compared to NC’s 213 days. This research sees the value of investigating the effectiveness of hands-on gardening to advance PA in the early years in varying conditions; especially in areas where FV gardening seemingly faces significant climatic/environmental challenges and obstacles.

Texas has the second (after New Mexico) highest concentration (39.1%) of the Hispanic and Latino population among all the states in the US. Among preschoolers (2–5 years), Latinos are three times more likely to be obese than Caucasians [12]. West Texas, with its semi-arid climate and a high percentage of the Hispanic and Latino population (37.1%), provides a unique site to investigate the effectiveness of hands-on FV gardening for advancing PA of children. The success (or failure) of gardening-based intervention for preventing obesity in early years in this region can provide valuable insights for similar interventions in other areas of the US.

The association between childcare attendance and risks of obesity in children [13,14,15] is of critical importance and demands special attention. Of the 21 million children in the US from birth to age 5 participating in various weekly non-parental care arrangements, 62% (more than 13 million in number) attend licensed childcare centers and spend most of their waking hours in those facilities [16]. Also, the most common location for children’s primary center-based care arrangement was a building of its own (42%) [16] where gardens can be installed conveniently. Preschool age, defined by the age range of 3–5 years [17], is a time when children start to learn following instructions, asking and answering simple questions, following rules or taking turns in games/lessons with other children, etc. [4]—making it an ideal age range for observing/experimenting the impacts of hands-on gardening on behavioral outcomes.

To examine the effectiveness of hands-on gardening intervention in licensed childcare centers in promoting preschoolers’ PA, this study conducted an experimental study with eight childcare centers and one hundred forty-nine children (*n* = 149) in Lubbock County located in WT. The study primarily aimed to investigate the association between childcare hands-on gardening and preschoolers’ PA in a semi-arid climate zone with a high percentage of Hispanic families and children. Whether a positive association (found in other age groups and different climate and demographic zones) extends to this study context may expand our understanding of hands-on gardening as a health intervention in early childhood environments.

## 2. Materials and Methods

### 2.1. Research Design

The proposed project adopts a research design of Randomized Two Group Pre- and Post-Test Experiment [18] as shown in Table 1.

The key concept behind this design comprises randomly assigning subjects (childcare centers) to two groups, an experimental, and a control group. Eight childcare centers were randomly assigned to two groups (four centers per group). Both groups were pre- and post-tested for children’s PA. However, the experimental group received the treatment—a hands-on FV gardening intervention. Randomization, here, was supposed to ensure that differences that might appear in the post-test would be the result of the experimental variable rather than the possible difference between the two groups to start with [18]. This classical type of experimental design has strong internal validity. The selection of this research design allowed the research team to compare the final post-test results, giving an idea of the overall effectiveness of the hands-on FV gardening intervention for children’s PA. The research team was able to analyze how both groups changed from pre-test to post-test and whether one, both, or neither performed differently in terms of child outcome measures on PA.

### 2.2. Participating Childcare Sites: Selection and Random Assignment to Groups

Licensed childcare centers vary greatly in many aspects including enrollment numbers, size, availability of outdoor areas, monthly fees, serving age ranges, quality ratings, permit types, childcare subsidies, etc. As independent businesses, childcare centers do not have the uniformity of public schools or preschools. Many of these aspects could result in biases and impact research outcomes. Therefore, a set of rigorous ‘selection criteria’ was established for enrolling childcare facilities in the study to ensure there were no ‘selection biases’ in the research design. The selection criteria for licensed childcare centers for the research included the following: (1) centers must be located in Lubbock County, West Texas (which falls within the semi-arid climate zone), (2) centers holding a full permit issued by the CCR (Child Care Regulation) of Texas Health and Human Services (THHS) during the study phase (to ensure compatibility of participating centers), (3) centers must enroll preschool-aged (3–5 years old) children and have separate classroom for preschoolers (for the feasibility of enrollment of child subjects and data collection), and (4) centers must accept childcare subsidies (to ensure children from low-income families will also have the opportunity to enroll in the research and participating centers are comparable in terms of family income). A list of 103 licensed childcare centers located in Lubbock County in West Texas was retrieved from THHS’s online childcare search tool [19] with the search criteria mentioned above. The list was scrutinized and further reduced to 83 eligible centers based on proximity and enrollment numbers. Driving distance from the university campus was critical as the research team needed to visit the centers several times for data collection and building the gardens. Centers were also scrutinized for their capacity (enrollment numbers) and low-capacity ones were excluded to ensure childcare centers enrolled in the study were comparable in sample size. With the help of the Texas Workforce Commission (TWC), an online call for applications was sent to the 83 eligible childcare centers. Only 13 completed applications were received out of those 83. Eight centers were randomly selected from those 13 and then randomly assigned to two groups—the experimental group with four childcare centers who received the hands-on garden intervention in year 1 (2022) and the control group with four childcare centers who did not receive the intervention in year 1 (2022) but received it in year 2 (2023). This arrangement ensured an experimental model in 2022 and an opportunity to compare pre- and post-intervention PA levels of children in the experimental (treatment) group and the control (non-treatment) group. The control group received the hands-on garden intervention a year later in 2023 to ensure child participants in those four centers were not deprived of opportunities to participate in gardening. The following diagram (Figure 1) shows how childcare centers and child subjects were recruited in the study.

### 2.3. Participating Children

In the selected eight childcare centers located in Lubbock County, Texas, parents received an invitation letter through the selected childcare center mail system to include their children in the study. The electronic and/or printed invitation letter included a brief description of the study and its potential benefits. Written consent forms (with one extra copy per child) were distributed to all parents in 3–5 years of age children’s classrooms in the participating centers by respective classroom teachers. Parents who agreed for their child(ren) to participate signed and returned the consent forms at their convenience in a designated box in the classrooms. The research team collected signed consent forms from the designated box with the permission of the classroom teacher. Children aged 3–5 years were only eligible to be enrolled in the research. A total of 185 children with parental approvals were initially enrolled in the study—93 children in the experimental group (E) centers and 92 in the control group (C) centers.

Valid ActiGraph data, however, were retrieved from a total of 149 children (*n* = 149)—81 children from the experimental group (E) centers and 68 from the control group (C) centers. ActiGraph data from 36 children (12 in the experimental group and 24 in the control group) were deemed unusable for various reasons described in the Limitations Section of the paper.

Among child participants, 53% were boys and 47% girls. The proportion of boys and girls in the experimental group (55% and 45%, respectively) was comparable with and similar to that of the control group (56% and 44%). The average age of child participants was 50 months with the experimental group (E) children, averaging 52 months, and it was 48 months for the control group (C) children during the start of the study. We were also interested in the participation (%) of Latino and/or Hispanic children (and families) in our study because of potential health disparities and obesity risks of Hispanic and Latino children as discussed earlier. As shown in Table 2, the percentages of participating Hispanic and/or Latino children in the experimental group (37%) and the control group (43%) were representative of Texas demographics of 39.1% of Hispanic and Latino population in general (second highest Hispanic and Latino concentration among all 50 states).

### 2.4. Variables

#### 2.4.1. Independent Variable: The Garden Intervention

The intervention for the experimental group (E) comprised a standardized FV garden component with six raised garden beds, designated FV cultivation, and children’s hands-on participation in FV gardening with their teachers guided by a Garden Activity Guide (tailored for the semi-arid climate zone) created by TTU’s partnering organization the Natural Learning Initiative (NLI) at NC State University. The guide introduces childcare professionals to a 12-step, hands-on learning process for preschool-aged children to engage them in seasonal FV gardening. Prior gardening experience of educators is not assumed. The guide provides a structured approach consisting of two main aspects of hands-on gardening—activities for the meaningful participation of children and the timing of harvesting varying the FV produce. The four FV gardens in the four experimental group (E) centers were built within the same time frame by the research team. The selection of FV included six vegetables, i.e., cucumbers, green beans, green peppers, tomatoes, yellow squash, and zucchini, and five fruits, i.e., blackberries, blueberries, cantaloupe, strawberries, and watermelon. The class teachers of participating preschool classrooms in the experimental group (E) led the implementation of the Garden Activity Guide with their children, which contained 12 gardening tasks suitable for the preschool age group (3–5 years old). The activities were categorized under three primary themes: preparation, maintenance, and harvesting/consumption with hands-on tasks ranging from inspecting seeds and readying garden plots to watering, weeding, and harvesting. Teachers led these activities with children for 30 min outdoor sessions, three times per week during the entire gardening season (late spring to early summer). The four control group (C) centers did not receive this intervention in year 1 of the research. They received the intervention in year 2. However, this paper is concerned with the data that we collected in year 1 comparing PA measures between the experimental group (E) and control group (C) children before and after the intervention, to investigate whether hands-on gardening contributed to a significant increase in PA levels of children.

#### 2.4.2. Dependent Variable: Physical Activity

Physical activity was assessed via the wActiSleep-BT (Actigraph Corp., Pensacola, FL, USA) accelerometer, which is a three-axis activity monitor worn on the wrist of the participant that provides a measure of the frequency, intensity, and duration of physical activity. The wActiSleep-BT is widely used in exercise and psychological sciences and has documented validity [20], and data collected using the wActiSleep-BT correlate highly with similar methods of measurement such as doubly labeled water (DLW) and indirect calorimetry [21,22]. Physical activity counts were recorded at a sample frequency of 60 Hz, at 1 s epochs, in order to increase measurement accuracy. Participants were asked to wear accelerometers on the wrist corresponding to their non-dominant hand for five days during the entire day (both waking hours and during sleep) and to return the accelerometers to study personnel after the 5-day time period. All participating children in both groups (experimental and control) attending the eight centers wore accelerometers on the same 5 days before and after the intervention (a total of 10 days). For each participant who wore the accelerometers, research staff recorded the dates and times that each accelerometer was given to the child and attached to their wrist, and when the accelerometer was removed. ActilifeTM 6 Software was utilized to assess the total wear time for each participant and to complete sedentary behavior, light activity, and moderate to vigorous physical activity (MVPA) estimates. Examples of sedentary behavior include activities such as sitting or standing stationary. Light activity includes walking, while MVPA includes any activity more than a walk (e.g., running and swimming; [23]). To estimate activity levels, the software bins moment-to-moment raw, triaxial acceleration values into counts per minute (CPM), which can then be used to classify sedentary behavior and physical activity based on various cut points [24]. In line with previous research [23], this study classified sedentary behavior as CPM of less than 3360. Light activity was defined as CPM between 3661 and 9804, and MVPA as CPM greater than 9805.

### 2.5. Data Collection Methods

The research team conducted several preoperational activities before data collection. The TTU research team visited the centers, met with the directors and teachers, and completed collecting all signed consent forms. Once all the consent forms were collected, the research team created separate lists of participating children for each of the eight participating centers. In those lists, all children were assigned unique random IDs. Stickers of those unique random IDs of children were attached to the accelerometer monitors. Unique random IDs ensured that the same accelerometer monitors were assigned to children during pre- and post-intervention data collection sessions. The unique random IDs also ensured data anonymity. Children started wearing the accelerometers on a Monday and wore them continuously till the Friday afternoon of that week. To initiate the data collection, a research team member will visit a preschool classroom of a participating childcare center on Monday morning at 8:30 am. The researcher carried all accelerometer monitors assigned with unique random IDs. He/she provided the classroom teacher with a list of children and their assigned unique random IDs. The list also had blank columns for recording times when children started wearing the accelerometers and when they took them off on Friday. The accelerometers were attached to the non-dominant wrist of each child with a static nylon belt by the trained researcher. The classroom teacher helped the researcher identify children by name and by attaching the assigned accelerometer monitors by matching the unique random IDs. The classroom teacher also recorded the times of taking accelerometers off on Friday afternoon. The researcher visited the center on the following Monday to collect all the accelerometers from the teacher. This process was repeated during the post-intervention data collection session. Teachers were compensated for each session for their help in the data collection process.

### 2.6. Statistical Methods

Repeated-measures analysis of variance (ANOVA) was performed using SPSS (IBM, 2023, IBM, Armonk, USA). The normality assumption was estimated by visually inspecting histograms and skewness statistics for levels of physical activity at time 1 and time 2. Histograms were observed to be relatively normally distributed, and all skewness statistics ranged between −1 and 1, indicating acceptably distributed data [25]. Further, outliers were evaluated using Cook’s distance. Cook’s distance values for all dependent variables ranged from 0.004 to 0.329, indicating there were no significant outliers [26]. Partial eta squared (*η*^2^) effect sizes were calculated for all times by group interactions. Qualitative descriptions for this effect size range from small (*η*^2^ = 0.01), to medium (*η*^2^ = 0.06), to large (*η*^2^ = 0.14; [27]. The statistical significance for all analyses was determined at *p* < 0.05.

## 3. Results

### 3.1. Statistical Analyses

All activity levels, i.e., sedentary, light, and MVPA, are presented as a percentage of valid wear time spent in each category to account for between-group differences in total wear time. Bivariate correlations were conducted to assess associations of age at Time 1 (before the hands-on gardening intervention), age at Time 2 (after the hands-on gardening intervention), and activity. It must be noted for clarity that the intervention (hands-on gardening) in this research was a continuous process, and during Time 2, children at the experimental group (E) centers were still actively participating in hands-on gardening activities with their teachers. Time 2 data, therefore, represents experimental group children’s PA during the intervention although it is referred to as post-intervention data in this paper. Repeated-measures three-way analyses of variance (ANOVAs) were performed to estimate interactions of group, time, and demographic variables (i.e., sex, age, and ethnicity) on activity. Repeated-measures two-way ANOVAs were performed to compare differences between group and time on activity throughout the total study duration, activity occurring only during school time, and activity occurring only outside of school time (e.g., at home, during other activities).

There were 24 participants with complete data at Time 1 and Time 2 for the intervention group and 17 with complete data at Time 1 and Time 2 for the control group. One center assigned to the control group did not allow children to take the ActiGraphs home after school. Thus, this group was only included in analyses assessing activity during school. To assess between-group differences between Hispanic and non-Hispanic participants (given the available sample size), participant ethnicity was dichotomized so that all individuals who did not identify as Hispanic were included in the non-Hispanic group.

### 3.2. Identification of Covariates

To estimate associations between age and PA, bivariate correlations were performed. As noted in Table 3, a significant and positive correlation was found between age at Time 1 (r = 0.23, *p* < 0.01) and light physical activity at Time 1 (r = 0.23, *p* < 0.01). Age at Time 2 (r = 0.26, *p* < 0.05) also demonstrated a significant and positive correlation with light physical activity at Time 2. Age at Time 1 (r = 0.42, *p* < 0.01) and Time 2 (r = 0.43, *p* < 0.01) was significantly and positively correlated with MVPA at Time 1. Thus, ages at Time 1 and Time 2 were included as covariates in subsequent models. 

As seen in Table 4, there was no significant interaction between gender, group, and percentage of time spent on sedentary activity (*p* = 0.30), light activity (*p* = 0.91), or MVPA (*p* = 0.53).

Further, as seen in Table 5, there were no between-group differences between Hispanic and non-Hispanic participants for sedentary behavior (*p* = 0.11), light physical activity (*p* = 0.11), and MVPA (*p* = 0.41).

### 3.3. Primary Analyses

Overall, there was no significant interaction between group and time on the percentage of sedentary behavior (*p* = 0.15), light activity (*p* = 0.24), or MVPA (*p* = 0.15) throughout the study duration (i.e., during school and out-of-school time, combined) while controlling for age at Time 1 and Time 2 (Table 6).

Regarding activity that only occurred during school time (Table 7 and Figure 2), there was a significant interaction between group and time on sedentary behavior (*F* = 4.33, *p* = 0.045, *η*^2^ = 0.110), controlling for age at Times 1 and 2. Pairwise comparisons were not significant. Specifically, neither the intervention group nor the control group participated in more sedentary behavior at Time 2 than at Time 1. However, comparing groups across time points can mask patterns of change because the difference across each time point is smaller than the net difference between the groups over time. Thus, a significant interaction is emphasized here.

There were no significant interactions of group and time with the percentage of sedentary behavior (*p* = 0.59), light activity (*p* = 0.63), or MVPA (*p* = 0.59) outside of school time (Table 8).

## 4. Discussion

This study compared preschoolers’ PA as an experimental outcome of their participation (or non-participation) in hands-on FV gardening activities while attending childcare centers in a semi-arid climate zone. The study also took a novel approach to investigate PA increases/changes as measured by accelerometers as a potential outcome of childcare hands-on gardening during childcare time (referred to as ‘school time’), home time, and the overall wear time (5 days continuously) referred to as ‘total time’. Analyzing data at these three levels makes this study unique compared to other relevant contemporary studies. The NC study [10] investigated similar effects only during school time (accelerometers were taken off before leaving childcare facilities), but it emphasized the need for investigating potential ‘spillover effects’ beyond school time at home. This study attempted to investigate this ‘spillover effect’ [28] by collecting PA data of children beyond school time while they were engaged in hands-on gardening at their respective childcare centers (experimental group). School gardens provide a dynamic outdoor environment where preschoolers can be physically active. Preschoolers can share their daily activities at childcare centers with their parents. Their enthusiasm about gardening, harvesting, and tasting FVs that they grew by themselves may be transmitted to their household/family culture of home gardening, outdoor times, and daily physical activity. There is no previous study that investigated spillover effects of hands-on gardening at preschool age (3 to 5 years). We did not find any significant differences in sedentary behavior, light physical activity, and MVPA between the experimental (gardening) and control (non-gardening) groups of children in home time and total time. However, this finding does not eliminate the need for investigating the ‘spillover effects’ of hands-on gardening on preschoolers’ PA. Our study was limited by a smaller sample size and ‘home time’ PA calculation was interrupted by one center not allowing its participating children to wear accelerometers at home—further reducing the sample size for a meaningful analysis.

The NC study reported a significant intervention effect for MVPA (*p* < 0.0001) and sedentary minutes (*p* = 0.0004), with children at intervention (gardening) centers acquiring approximately 6 min more MVPA and 14 min less sedentary time each day while attending their respective childcare centers (school time) [11]. We did not find any significant differences (*p* = 0.15) between the experimental (gardening) and the control (non-gardening) group (Table 7 and Figure 2). Both groups experienced an increase in their MVPA (%). The positive association indicates the experimental group enjoyed a higher increase in their MVPA but not significantly more than the control group. However, like the NC study, we also found a significant reduction in sedentary time (*p* = 0.045) for the experimental group of children who participated in hands-on gardening. The only difference was observed for the significance levels—the NC study found the difference to be highly significant (*p* = 0.0004) while our study found a moderately significant difference at *p* ≤ 0.05 level. This similar finding for the reduction in children’s sedentary behaviors as an experimental outcome of hands-on gardening is meaningful in many ways. Diverse opportunities for PA are important in childcare centers, especially in outdoor environments. Children’s PA requirements, interests, and motivations can be different and a childcare outdoor environment with diverse PA affordances can ensure that children are engaged in moderate/light to vigorous PAs. Hands-on gardening brings a variety of PA opportunities for children and adds to diversified PA affordances for preschoolers. A significant reduction in sedentary time outdoors supports this hypothesis in both the NC study (in a humid subtropical climate zone) and the Texas one (in a semi-arid climate zone). If hands-on gardening can contribute to reduced sedentary behaviors of preschoolers in their outdoor times while attending childcare centers in two drastically different climate zones, it should be considered a nationally implementable early health intervention for preventing childhood obesity.

Our other important finding, though not highly significant in statistical terms, implies the value of adding hands-on gardening in childcare outdoor activities to increase the light activity of children. The NC study reported no significant differences in light physical activity between the two groups of children—the experimental (with gardening intervention) vs. the control (no gardening intervention) groups. Our ‘school time’ analyses (Table 7 and Figure 2) show that children exposed to hands-on gardening experienced an increase in their light activity compared to the non-gardening group. This difference is statistically significant at *p* ≤ 0.1 (*p* = 0.078). This denotes a shred of weak evidence or trend, but given our small sample size (*n* = 149), this difference demands in-depth discussion.

To understand this finding, we investigated the list of activities defined by the 5-level Children’s Activity Rating Scale (CARS) [29] under Level 2 (low; easy) and Level 3 (medium; moderate). Movement behaviors that are closely associated with gardening fall mostly under Level 2 and Level 3 (Table 9). Since the coding of the CARS scale is validated by accelerometry cut points [30], this observation of a significant (*p* ≤ 0.1) increase in light physical activity due to participation in hands-on gardening is meaningful. Accelerometers can measure the PA levels but cannot predict the nature/type of PAs children were involved in. Future research comparing accelerometry data and observational data of preschoolers’ PA while engaging in hands-on gardening will be valuable to obtain a more in-depth understanding of gardening-based PA.

One other key objective of the study was to investigate whether changes in PA as an outcome of hands-on gardening were significantly different between Hispanic vs. non-Hispanic preschoolers. We did not find any statistically significant differences between the two groups. The health disparities and higher risks of childhood obesity in Hispanic children inspired this investigation. But we acknowledge that our findings due to a limited sample size may need further validation Thus, more studies with larger sample sizes are needed.

## 5. Limitations

The challenges and the limitations of the study are equally important discussions that shed light on many realities of intervention research involving preschoolers attending childcare centers. We repeatedly mention our limitation related to a small sample size, but this limitation is caused by a complex set of reasons.

First, we faced challenges in recruiting childcare centers for this study. Although we built and managed the FV gardens in the experimental group (E) centers and provided participation support costs to all teachers for each data collection session, very few centers were fully on board with participation. Childcare centers are consistently challenged by teacher turnover and limited resources. Texas has one of the highest turnover rates in early childhood education at over 20% [31]. Even when childcare center leadership (owner/director) is enthusiastic to implement hands-on gardening in their outdoor environment, they are overwhelmed by any ‘extra effort’ while constantly navigating through the challenges of finding the personnel to run their enterprise. Hands-on gardening as a health intervention is long-term and complex. For tracking data related to the intervention (children’s participation in hands-on gardening), we had to depend on classroom teachers. Although they were compensated for their participation and efforts, numerous times our study team had to adjust to classroom teachers leaving the childcare center and reassign new teachers in the study.

Second, children and families leaving their childcare centers created challenges, especially for collecting data in the post-intervention phase. In this phase of data collection after the garden intervention, it was documented that there was a 12% loss of sample size in the experimental group due to non-continuation. Likewise, the control group had a loss of 8% of their sample data for the same reasons—children leaving or graduating from their respective childcare centers.

Third, during each data collection, the children were asked to wear accelerometers continuously for five consecutive days. In the recorded final accelerometer data, 12% of the sample data from the experimental group and 4% from the control group were lost due to missing data resulting from non-wear (i.e., children refusing to wear the accelerometers for an extended period, or they simply lost the devices in their home environments). This showed how challenging it is to collect beyond-school PA data of preschoolers attending childcare centers. One of the authors (J.V.A.) of this study has extensive experience of using accelerometers for collecting continuous PA data of school-agers and never experienced data and device losses during this study. It indicates that alternative methods are needed for collecting continuous PA data of younger children.

Although we were able to recruit 185 children for the study (received signed consent forms from parents), due to the three reasons noted above, data from only 145 children were usable for statistical analyses of overall interactions. For pairwise comparisons, this sample size was further reduced to only 41, limiting our ability to conduct any meaningful analyses at the individual level, Specifically, missing data that resulted in this smaller sample size reduced the ability to identify significant interactions if any were present. This may explain why we were only able to detect the largest effect, which was the interaction of group and time with sedentary behavior. A larger sample size would provide greater confidence in study results, while also contributing to greater generalizability and additional options with regard to nuanced data analyses that include more moderators and potential mechanisms (e.g., mediators) of the associations examined in this study.

## 6. Conclusions

In the changing landscape of the financial reality of licensed childcare centers in the US, we must find low-cost, sustainable, and feasible health interventions to ensure the health and well-being of young children. A hands-on FV gardening component (found to be successful even in a semi-arid climate zone with relatively low rainfall and humidity) should be considered a feasible health intervention applicable to most US childcare centers. While accelerometer data captured PA levels of children essential for understanding PA variations as an outcome of hands-on gardening intervention for preschoolers, future research should attempt to achieve a deeper understanding of PA types. Our observational data (not included in this paper) showed that children engaged in activities such as carrying, lifting, kneeling (or squatting), digging, etc. when they actively participated in hands-on gardening tasks like watering, seeding, weeding, and harvesting. These PA behaviors potentially could mediate a PA-level increase for the intervention group. However, accelerometer data can capture only PA levels and not PA diversity or types. Future research on the topic of gardening-based health intervention for preschoolers should include observational data (e.g., video data and behavior mapping data) for a deeper understanding of children’s PA types and diversity associated with hands-on gardening at childcare centers.

## Figures and Tables

**Figure 1 ijerph-21-00548-f001:**
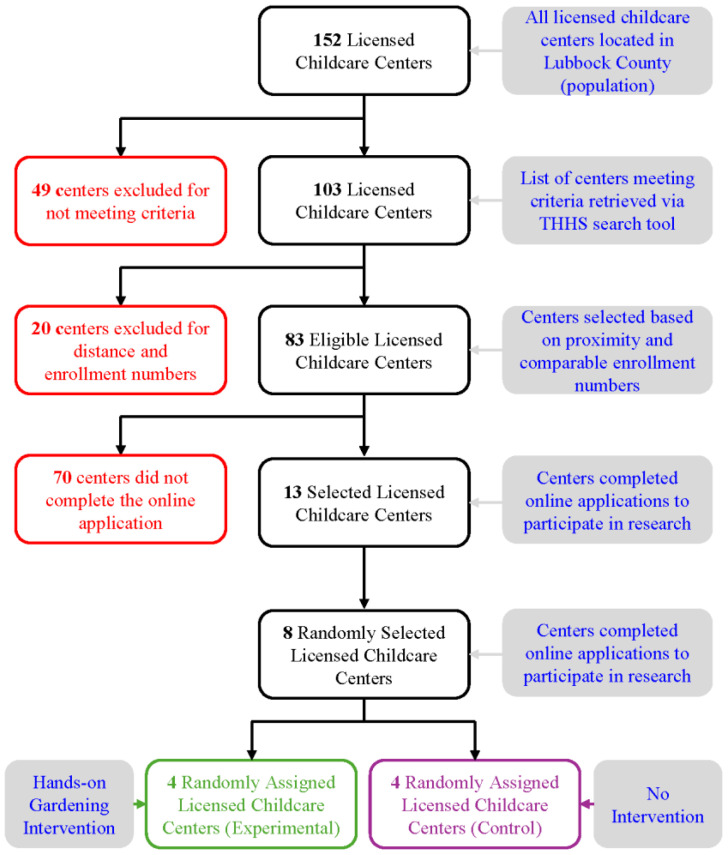
Flow diagram of childcare center selection process.

**Figure 2 ijerph-21-00548-f002:**
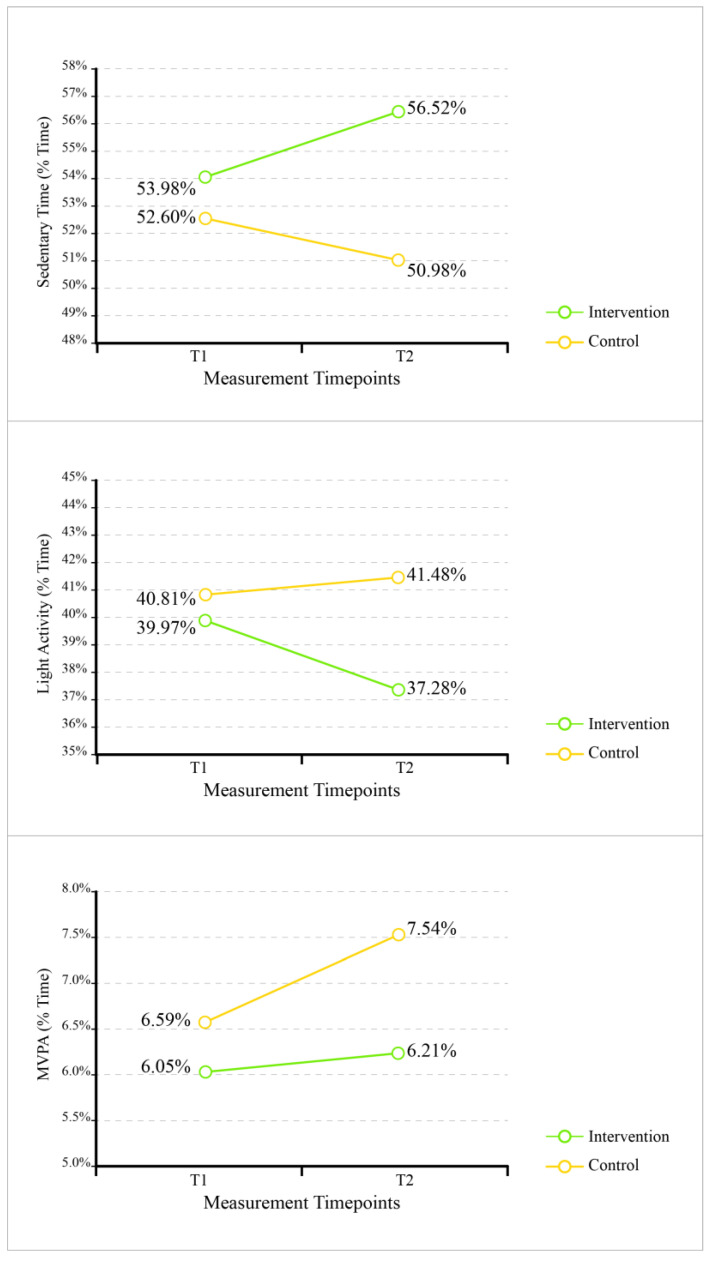
Percentage of in-school time in each category of sedentary behavior, light physical activity, and moderate to vigorous physical activity.

**Table 1 ijerph-21-00548-t001:** Two Group Pre- and Post-Test Experiment.

Group	Pre-Test	Treatment	Post-Test
Experimental group = E	O_1_	X	O_2_
Control group = C	O_1_		O_2_

Experimental group: E = 4 childcare centers with 81 child participants. Control group: C = 4 childcare centers with 68 child participants. X = standardized six-raised-bed FV hands-on garden intervention in summer. O_1_ = pre-test data collection in spring (before garden intervention) on children’s PA using accelerometers. O_2_ = post-test data collection in fall (after garden intervention) on children’s PA and FV preferences.

**Table 2 ijerph-21-00548-t002:** Ethnic distribution of participants.

Ethnicity	Experimental Group (E)	Control Group (C)
Non-Hispanic White or Euro-American	41%	30%
Black, Afro-Caribbean, or African American	3%	8%
Latino and/or Hispanic American	37%	43%
East Asian or Asian American	5%	0%
South Asian or Indian American	7%	11%
Middle Eastern or Arab American	2%	0%
Multi-racial	1%	2%
Other	4%	6%

**Table 3 ijerph-21-00548-t003:** Correlations between outcomes and age.

	Sed. (1)	Light (1)	MVPA (1)	Sed. (2)	Light (2)	MVPA (2)	Age (1)	Age (2)
Sed. (1)	1	0.75 **	0.41 **	0.07	−0.01	−0.02	0.04	0.07
Light (1)		1	0.70 **	0.03	0.04	0.08	0.23 **	0.24 **
MVPA (1)			1	−0.23	−0.34 *	−0.17	0.42 **	0.43 **
Sed. (2)				1	0.80 **	0.60 **	−0.09	0.13
Light (2)					1	0.77 **	0.04	0.26 *
MVPA (2)						1	0.06	0.25 *
Age (1)							1	0.97 **
Age (2)								1

Sed. = sedentary behavior, Light = light physical activity, and MVPA = moderate to vigorous physical activity. *Note:* the number in parenthesis indicates outcome or age at either time one (1) or time two (2). *** Correlation is significant at *p* < 0.05. ** Correlation is significant at *p* < 0.01.

**Table 4 ijerph-21-00548-t004:** Outcomes for gender.

Outcomes	Gender	Group	Time 1	Time 2	*F*	*p*	*η* ^2^
Sedentary	Male	Intervention	49.22%	54.70%	1.09	0.304	0.03
		Control	58.39%	60.04%			
	Female	Intervention	56.77%	51.46%			
		Control	55.08%	54.33%			
Light	Male	Intervention	43.85%	38.42%	0.013	0.911	0.000
		Control	36.98%	34.64%			
	Female	Intervention	37.39%	41.91%			
		Control	37.37%	39.37%			
MVPA	Male	Intervention	6.94%	6.88%	0.398	0.532	0.012
		Control	4.64%	5.32%			
	Female	Intervention	5.84%	6.63%			
		Control	7.55%	6.30%			

Sedentary = sedentary behavior, Light = light physical activity, and MVPA = moderate to vigorous physical activity.

**Table 5 ijerph-21-00548-t005:** Outcomes for ethnicity.

Outcomes	Ethnicity	Group	Time 1	Time 2	*F*	*p*	*η* ^2^
Sedentary	Hispanic	Intervention	55.86%	53.34%	2.68	0.110	0.075
		Control	58.89%	60.30%			
	Non-Hispanic	Intervention	52.04%	52.99%			
		Control	55.47%	55.53%			
Light	Hispanic	Intervention	39.30%	40.64%	2.635	0.114	0.074
		Control	36.54%	34.80%			
	Non-Hispanic	Intervention	41.06%	40.01%			
		Control	37.66%	38.05%			
MVPA	Hispanic	Intervention	4.84%	6.02%	0.706	0.407	0.021
		Control	4.57%	4.91%			
	Non-Hispanic	Intervention	6.91%	7.00%			
		Control	6.88%	6.42%			

Sedentary = sedentary behavior, Light = light physical activity, and MVPA = moderate to vigorous physical activity.

**Table 6 ijerph-21-00548-t006:** Percentage of time in each category throughout the entire study duration.

Outcomes	Group	Time 1	Time 2	*F*	*p*	*η* ^2^
Sedentary	Intervention	52.99%	53.08%	2.195	0.147	0.059
	Control	57.06%	57.76%			
Light	Intervention	40.62%	40.17%	1.435	0.239	0.039
	Control	37.14%	36.53%			
MVPA	Intervention	6.39%	6.75%	2.140	0.152	0.058
	Control	5.80%	5.71%			

Sedentary = sedentary behavior, Light = light physical activity, and MVPA = moderate to vigorous physical activity.

**Table 7 ijerph-21-00548-t007:** Percentage of in-school time in each category.

Outcomes	Group	Time 1	Time 2	*F*	*p*	*η* ^2^
Sedentary	Intervention	52.60%	50.98%	4.330	0.045 *	0.110
	Control	53.98%	56.52%			
Light	Intervention	40.81%	41.48%	3.290	0.078	0.086
	Control	39.97%	37.28%			
MVPA	Intervention	6.59%	7.54%	2.154	0.151	0.058
	Control	6.05%	6.21%			

* Correlation is significant at *p* < 0.05. Sedentary = sedentary behavior, Light = light physical activity, and MVPA = moderate to vigorous physical activity.

**Table 8 ijerph-21-00548-t008:** Percentages of out-of-school time in each category.

Outcomes	Group	Time 1	Time 2	*F*	*p*	*η* ^2^
Sedentary	Intervention	54.91%	58.60%	0.304	0.586	0.011
	Control	62.01%	61.96%			
Light	Intervention	39.67%	36.59%	0.242	0.627	0.009
	Control	33.99%	33.87%			
MVPA	Intervention	5.42%	4.81%	0.292	0.593	0.011
	Control	4.01%	4.18%			

Sedentary = sedentary behavior, Light = light physical activity, and MVPA = moderate to vigorous physical activity.

**Table 9 ijerph-21-00548-t009:** Preschoolers’ PA Levels (Level 2 and 3) as defined by the Children’s Activity Rating Scale (CARS).

Activities	Association with Gardening	CARS Levels
Standing, sitting, squatting, kneeling, digging	Most gardening activities such as watering, seeding, weeding, harvesting, etc.	Level 2 (low; easy)
Walking at a leisurely pace, crawling, sit-ups	Some gardening activities like watering, observing, harvesting, etc.	Level 3 (medium; moderate)

## Data Availability

The data presented in this study are available upon request from the corresponding author (M.M.).

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
