# Peer review of "Effects of Childcare Hands-On Gardening on Preschoolers’ (3–5 Years) Physical Activity in Semi-Arid Climate Zone"

_ijerph, 2024, doi:10.3390/ijerph21050548_

Round 1
Reviewer 1 Report
Comments and Suggestions for Authors
The article comprises a relevant issue and was accordingly designed, however, there are some relevant points to be observed regarding text presentation and organization, and statistics. Below, such points are detailed:
At the aim, instead “corroborate findings” of other study, would be better if just “investigate the association…”. Eventually, the discussion can conduct some comparisons with other studies to see if your findings can corroborate findings of other study.
Statistical Analyses fits better at the end of methods section. Please, insert at article’s text the criteria used to elicit the parametric stats, informing if such criteria were achieved. Also inform at text the significance level and used software.
It is necessary to objectively define at article’s text (methods section) the activity levels (i.e., sedentary, light, and moderate to vigorous or MVPA). How it is measured? Inform the meaning of MVPA at the first time it appears.
Line 52: inform the meaning of “FV”.
Regarding all abbreviations along with the article, is necessary a revision. Please include the full text at the first time it appears. If I am not wrong, FV is only fully wrote at conclusion, but used at first time at introduction’ second paragraph. At table 2, for example, it is not informing the meaning of “Sed.” (despite the reader can imagine the meaning, it is not enough). Table 3 not inform the meaning of MVPA. Please carefully review all article’s text about it.
Figures 2, 3 and 4 could be inserted in only one figure with panel A, B and C. Graph of figure 4 could be axis Y scale adjusted to initiate at 5% (or 4%).
The real study conclusion is only lines 438 to 441. The rest of the conclusion’s text must be removed or inserted in another section of the article.
Reviewer 2 Report
Comments and Suggestions for Authors
The introduction provides a substantial background on the importance of preventing obesity from an early age and the potential role of hands-on gardening in promoting physical activity among preschoolers. It effectively contextualizes the study within a broader research landscape, citing relevant studies and statistics to justify the research focus. The cited references are relevant and contribute to establishing a solid foundation for this study. However, the inclusion of a more detailed discussion on the specific challenges and opportunities presented by semi-arid climates for gardening interventions could further enrich the background.
The cited references are relevant and provide a strong theoretical and empirical foundation for this study. They effectively span across related domains, such as obesity prevention, the impact of gardening on physical activity, and early childhood development.
The randomized two-group pre- and post-test experimental design is appropriate and robust for investigating the effects of the gardening intervention on preschoolers’ physical activity. This design allows for the control of potential confounding variables and ensures that differences in physical activity outcomes can be attributed to the intervention.
The Methods section is comprehensive and provides a detailed description of the research design, participant selection, garden intervention, and data collection procedures. However, more clarity on the selection criteria for childcare centers and the specifics of the garden activity guide tailored for semi-arid climate zones would enhance the reproducibility of the study.
The results are presented with appropriate statistical analyses to assess the impact of gardening intervention on physical activity levels. The use of tables and figures enhances the presentation and understanding of the findings. However, a deeper analysis of the data, considering demographic variations among participants, could provide more nuanced insights.
The conclusions drawn from the study are well supported by the results. The study found significant effects of the gardening intervention on reducing sedentary behavior and emphasized the potential of hands-on gardening as an obesity prevention strategy. These conclusions align with the documented outcomes and contribute valuable knowledge to the field.
The document is well written with a clear and concise use of the English language. There were no apparent issues with plagiarism.
There were no evident inappropriate self-citations. These citations appear to be relevant and necessary for the context of the study.
No ethical concerns were evident from the review. The study was conducted in accordance with ethical standards, and informed consent was obtained from all subjects involved in the study.
Accept after minor revisions (corrections to minor methodological errors and text editing).
This study presents valuable insights into the effects of hands-on gardening on physical activity among preschoolers in a semi-arid climate zone. With minor revisions for additional clarity in methods and a deeper analysis considering demographic variations, this study would be a strong contribution to the field.
1. Enhancements to the methods section for clarity
- Detailed description of garden activity guide: Provide a more comprehensive description of the Garden Activity Guide tailored for semi-arid climate zones. Specifically, details of the activities included how they were adapted to address the challenges of gardening in a semi-arid climate, and any modifications made to suit the developmental stages of preschoolers. This addition will help readers to understand the applicability and potential replicability of the intervention in similar climates.
- Clarification on Childcare Center Selection Criteria: Expand the selection criteria for childcare centers, especially how they were evaluated for suitability for gardening intervention. Specific characteristics of the semiarid climate that influenced the selection and how these factors were expected to impact the study's outcomes were included.
2. Deeper analysis of demographic variations
- Subgroup analyses: Conduct and present analyses that explore how the intervention's effects may vary across different demographic groups, such as age, gender, and ethnicity. Given the diverse population in the study area, understanding how these factors influence the effectiveness of gardening interventions is crucial for tailoring future programs.
- Impact of the socioeconomic status of the participants was considered as a factor in the analysis. Since gardening interventions require resources and support, understanding how SES might influence outcomes can provide insights into accessibility and equity issues.
3. Minor methodological corrections
- Data collection consistency: Address the issue of data loss due to accelerometer non-wear and device loss, which affected the sample size for statistical analyses. Future studies should explore alternative methods for continuous physical activity data collection in young children, or implement strategies to minimize data loss, such as increased monitoring and incentives for consistent device wear.
- Recruitment challenges: Discuss the challenges encountered during the recruitment of childcare centers and families. Provide recommendations for engaging centers and parents in similar future research, potentially through enhanced communication strategies or highlighting the benefits of participation.
4. Text editing
- Wherever possible, simplify technical terms and statistical language to make the manuscript more accessible to a broader audience, including practitioners and educators in the early childhood and public health sectors.
- Ensure consistent use of terms throughout the manuscript, especially when referring to the study design, intervention activities, and outcome measures.
Comments on the Quality of English LanguageThe document is well written with a clear and concise use of the English language.
Reviewer 3 Report
Comments and Suggestions for Authors
Dear Authors,
I am honored to have the opportunity to review your insightful study, "ijerph-2952045," which examines the influence of hands-on gardening activities on the physical activity levels of preschoolers (aged 3-5 years) in childcare centers within a semi-arid climate. This research is pivotal in assessing whether practical gardening interventions can enhance physical activity and potentially aid in obesity prevention in children, considering the unique environmental challenges of a semi-arid climate.
Your manuscript is a fitting contribution to the "Exercise and Health-Related Quality of Life" section of the Special Issue on “Exercise in Living Environments: A Healthy Lifestyle.” The study stands out for its originality, practical relevance, and its potential to significantly contribute to obesity prevention strategies, underscored by a robust experimental design and solid statistical analysis.
However, to enhance the clarity and impact of your study, I recommend certain refinements:
1. English Language Enhancements: The manuscript's language is generally clear and well-structured. However, a few minor grammatical and typographical corrections are needed for improved clarity:
1.1. Please revise line 186 to correct the term "bentween."
1.2. In line 366, there appears to be an extraneous letter “p.”
1.3. In Section 2.4.1. Independent Variable: The Garden Intervention, consider revising "for a 30 minutes outdoor session three times per weekduring the entire gardening season" to "for 30-minute outdoor sessions three times per week during the entire gardening season" for greater accuracy and readability.
1.4. In the 3.1. Statistical Analyses section, "It most be noted for clarity that the intervention" should be amended to "It must be noted for clarity that the intervention."
2. Activity Categorization: Could the study further categorize the types of activities children engage in during the gardening sessions, beyond measuring their intensity?
3. Limitations and Discussion Enhancement: While limitations are acknowledged, a deeper discussion on potential biases or confounding factors would enrich the study’s interpretive value. Specifically:
3.1. Explicate how these limitations might influence the study's results.
3.2. Focus the discussion on these specific limitations to provide a clearer understanding of their impact on the findings.
I am confident that addressing these points will significantly refine and strengthen your manuscript. I eagerly await the revised version of your research and its valuable contribution to the field.
Comments on the Quality of English LanguageThe overall use of English in the manuscript is commendable. However, to enhance readability, I suggest fine-tuning some sentence structures. This will improve the clarity and fluency of the language, making the paper more coherent for readers. Detailed suggestions have been provided in the comments to the authors. Additionally, there is a blank page within the document that should be removed to maintain a professional presentation.
Round 2
Reviewer 1 Report
Comments and Suggestions for Authors
I appreciate the authors’ effort to solve the raised questions. Despite some of them (as my comment regarding discussion section) were not fully responded, I see that it is not enough to deny the publication. I also highlight my view that the conclusion is too long, but again, it is more about the writing style then something mandatory of changing for publication. However, I recommended a careful revision of the text presentation once there are many text editing errors. Some of them (not all) are cited below:
Please, review the tables’ number and its citation at the text.
At line 308 there is a table (should be table 3). Again, was not informed about the meaning of “Sed.”. Please, carefully observe the tables, all of them. The table 4, 5, 6 and 7 have no legend to identify MVPA for example. At table 8 (and text), what is the meaning of CARS? Tables and figures must insert information enough for its interpretation without the need for the reader to go to the article’s text.
The resolution of figure 2 can be improved.
Reviewer 3 Report
Comments and Suggestions for Authors
Dear Authors,
I express my sincere gratitude for reviewing the new version of your study. The collective efforts from you and your colleagues are reflected in the significant improvements made to Manuscript ID 'ijerph-2952045.'
I commend you for diligently addressing all previous comments and meeting the outlined requests, thereby significantly enhancing the overall quality of the research. Having carefully reviewed the authors' response letter, I agree with the improvements made in this second version.
I have no additional questions and appreciate the opportunity to contribute to the review process and eagerly anticipate witnessing the continued development of your manuscript.
Thank you once again for your dedication to advancing scientific knowledge.
